# Synthesis and Characterization of Amine and Aldehyde-Containing Copolymers for Enzymatic Crosslinking of Gelatine

**DOI:** 10.3390/ijms25052897

**Published:** 2024-03-01

**Authors:** Silvana Alfei, Federica Pintaudi, Guendalina Zuccari

**Affiliations:** Department of Pharmacy (DIFAR), University of Genoa, Viale Cembrano, 4, 16148 Genoa, Italy; fede.pinta85@gmail.com (F.P.); guendalina.zuccari@difar.unige.it (G.Z.)

**Keywords:** tissue engineering (TE), gelatine crosslinking, amino butyl styrene copolymers, acryloyl amidoamine copolymers, imine bonds, aldolic reaction, gelatine crosslinking, lysyl oxidase enzyme (LO)

## Abstract

In tissue engineering (TE), the support structure (scaffold) plays a key role necessary for cell adhesion and proliferation. The protein constituents of the extracellular matrix (ECM), such as collagen, its derivative gelatine, and elastin, are the most attractive materials as possible scaffolds. To improve the modest mechanical properties of gelatine, a strategy consists of crosslinking it, as naturally occurs for collagen, which is stiffened by the oxidative action of lysyl oxidase (LO). Here, a novel protocol to crosslink gelatine has been developed, not using the commonly employed crosslinkers, but based on the formation of imine bonds or on aldolic condensation reactions occurring between gelatine and properly synthesized copolymers containing amine residues via LO-mediated oxidation. Particularly, we first synthesized and characterized an amino butyl styrene monomer (**5**), its copolymers with dimethylacrylamide (DMAA), and its terpolymer with DMAA and acrylic acid (AA). Three acryloyl amidoamine monomers (**11a–c**) and their copolymers with DMAA were then prepared. A methacrolein (MA)/DMAA copolymer already possessing the needed aldehyde groups was finally developed to investigate the relevance of LO in the crosslinking process. Oxidation tests of amine copolymers with LO were performed to identify the best substrates to be used in experiments of gelatine reticulation. Copolymers obtained with **5**, **11b**, and **11c** were excellent substrates for LO and were employed with MA/DMAA copolymers in gelatine crosslinking tests in different conditions. Among the amine-containing copolymers, that obtained with **5** (CP5/DMMA-43.1) afforded a material (M21) with the highest crosslinking percentage (71%). Cytotoxicity experiments carried out on two cell lines (IMR-32 and SH SY5Y) with the analogous (P5) of the synthetic constituent of M21 (CP5/DMAA) had evidenced no significant reduction in cell viability, but proliferation promotion, thus establishing the biocompatibility of M21 and the possibility to develop it as a new scaffold for TE, upon further investigations.

## 1. Introduction

Tissue engineering (TE) is a modern research field that deals with the replacement, repair, or regrowth of damaged or malfunctioning tissues and organs [1,2,3,4,5,6,7]. The first works on the topic appeared in the seventies, but the birth of the term tissue engineering can be traced back to 1988 at a workshop held in February in Granlibakken, Lake Tahoe (CA, USA) [8]. Later (1993), the work in which TE was defined as “an interdisciplinary field in which the principles of engineering and life sciences are applied” was published [9]. The crucial phases of TE consist of extracting cells from the tissue of the donor patient, cultivating the extracted cells in vitro, making them proliferate on a support structure (scaffold), and implanting the new tissue in the donor patient via surgery or injection (Figure 1). Ideally, the scaffold should resist after implantation for the time necessary for the growth of the new tissue and then be replaced by it.

The application potential of TE is enormous due to the great disproportion existing on a global scale between the number of patients requiring transplantation and the number of organs available. Moreover, enormous is the variety of organs such as the liver, kidneys, and pancreas, or tissues such as blood vessels, skin, cartilage, bones, ligaments, and tendons, for which it would be very useful to have effective substitutes. From the early successes of TE, such as the introduction of long-standing commercial skin substitutes [10] and the construction of bladders [11], to the more recent achievements, including the replacement and repair of cardiac or nervous tissue [12,13,14], the repair of bone defects [14,15], and the bone and dental bone regeneration [16], progress has been massive both in terms of scaffold characteristics and cell sources, nowadays including also stem cells [6]. Scaffolds play a pivotal role in TE, being essential both for providing mechanical support to the cells that are forming the new tissue and for defining the dimensions and shape of the tissue itself. The scaffolds can be implanted in the patient once “seeded” with the cells of interest to allow the tissue to develop in vivo. Otherwise, the scaffolds can be used for cell proliferation in vitro using suitable reactors until a material with characteristics close to those of the desired tissue is obtained and then implanted. The ideal requirements for good support to be used in TE include biocompatibility without inducing immune responses (inflammation, mainly) and proper mechanical properties to guarantee the maintenance of the three-dimensional (3D) structure required during the growth of the tissue and its mouldability. A good scaffold should be biodegradable after implantation without the production of toxic residues and capable of carrying out those typical activities of the extracellular matrix (ECM), such as promoting cell adhesion and allowing the diffusion of nutrients, metabolites, and growth factors. Additionally, the surface properties, porosity, pore dimensions, and pore distribution are of paramount importance in support of regenerative medicine. Different types of materials have been used as support structures for TE. Inorganic materials such as metals [17] and ceramics [18,19] have found applications in the preparation of bone tissue substitutes but are poorly appropriate for other types of tissues. Both natural and synthetic polymers [20,21] have proven to be more versatile due to their chemical and mechanical properties closer to those of most of the natural tissue. Synthetic polymers used in TE can be classified as non-biodegradable and biodegradable [20,21,22]. Non-biodegradable polymers such as polymethyl methacrylate (PMMA) and poly-hydroxyethyl methacrylate (PHEMA) have contraindications and provide undesired effects, including inflammation, mainly in long-term implants. It has been reported, in fact, that in bone regenerative therapy, the long-term presence of non-biodegradable scaffolds caused impaired bone formation, difficult radiological assessment of bone healing, and inflammation. Additionally, non-biodegradable scaffolds loaded with antibiotics to prevent bone infections showed slow and prolonged drug release [23]. It has been demonstrated that non-biodegradable PMMA scaffolds, due to a prolonged and low release of antibiotics, caused the emergence of bacterial resistance, and gentamicin-resistant staphylococci were recovered from the surface of PMMA beads [24]. Otherwise, biodegradable polymers, including polyglycolic acid (PGA), polylactic acid (PLA), polyurethane (PU) as well as copolymers of lactic and glycolic acids (PLGA), are more attractive materials. Although not completely removed, they can be broken down in vivo to produce macromolecular degradation products [25]. In this context, polyphosphazenes are a promising class of polymers containing nitrogen and phosphorus atoms, which can be functionalized with a large variety of substituents, thus providing biodegradable biomaterials [26]. Particular types of biodegradable materials are even more interesting, such as bio-resorbable polymers (poly-D,L-lactide), which are solid polymeric materials undergoing bulk degradation and fully resorption in vivo [25]. Moreover, bio-erodible polymers, including poly-(α-hydroxy) acids, crosslinked polyester hydrogels, poly(ortho-esters), and polyanhydrides, are solid polymeric materials that undergo surface degradation (erosion) that continue uninterrupted until the material is fully eliminated [25].

Also, bio-absorbable polymers, such as polyethylene glycol (PEG), are solid polymeric materials that can slowly dissolve in body fluids without any cleavage of the polymer chain or any decrease in the molecular mass [25].

However, natural polymers such as protein-based materials composing the ECM, including collagen, its derivative gelatine, and elastin, as well as polysaccharides, including hyaluronic acid (HA), which are capable of forming 3D hydrogels, have attracted the attention of researchers as possible scaffolds for TE [27]. Natural polymers were used to fabricate biomaterials to assist tissues such as bone, bladder, cartilage, corneas, heart valves, nerves, pancreas, and skin [14]. Collagen has been used in the reconstruction of bladders [11] as a substitute for skin [28], nervous tissues [29], and small intestine [30]. Anyway, collagen is an expensive material that can induce immunogenic responses. Otherwise, gelatine is a low-cost protein material that does not exist in nature and derives from collagen by denaturation of the triple helix structure into single chains upon acid or basic treatment followed by heating. Acid treatment provides type A gelatine, while basic treatment gives type B gelatine [31]. Gelatine can form hydrogels that have been used in the vascularization of engineered tissues without causing immunogenic responses. Additionally, due to its numerous properties, including biodegradability and biocompatibility, gelatine is used in various sectors. Specifically, in the biomedical field, gelatine has proven to be an excellent vehicle for the controlled release of bioactive molecules [32]. Unfortunately, scaffolds made only of collagen- or gelatine have been demonstrated to not possess adequate mechanical properties (modulus, strength), including stiffness and biodegradation rate, for in vitro handling and long-term cell culture [33,34]. In nature, collagen is stabilized and reinforced by reticulation through the lysyl oxidase (LO)-mediated oxidation of the ε-amino groups of its lysine and hydroxylysine moieties to aldehyde groups and subsequent aldolic condensations reactions or imine bonds formation between different chains according to Figure 1.

In this regard, researchers thought to improve the mechanical properties of collagen and gelatine by using crosslinking processes capable of forming chemical bonds between its different chains as occurs in nature for collagen. Different crosslinking agents, including glutaraldehyde [35] and 1-ethyl-3-(3-dimethylaminopropyl) carbodiimide hydrochloride (EDC), have been used to crosslink collagen [25,26,36].

Concerning gelatine, the most adopted crosslinking agents include formaldehyde [37] and glutaraldehyde [38,39,40] in combination or not with resorcinol, genipin [38,39,41], ethylene glycol diglycyl ether [39,42], lysine diisocyanate [43,44], poly-aldehydes obtained from partially oxidized natural polysaccharides [45,46,47], soluble carbodiimides [39,41] and more recently biodegradable polymers. Additionally, enzymes such as transglutaminase or tyrosinase have been introduced to promote the crosslinking of gelatines [48]. Against this background, the main scope of this work was to develop a novel and versatile protocol to crosslink gelatine and potentially afford mechanically improved materials with possible application as a scaffold for TE. So, starting from the natural example of collagen reticulation by LO-mediated oxidation described in Figure 1, we considered crosslinking gelatin using LO in the presence of polymers properly designed and synthesized by us. To this end, we synthesized amine-containing hydrophilic copolymers and a terpolymer, mimicking the lysine residues of collagen (Figure 1) as possible substrates for LO. The enzymatic oxidation of amine groups of the prepared polymers by LO to aldehyde ones would have promoted the reticulation processes shown in Figure 1, as exposed in Figure 2 and Figure 3.

A methacrolein (MA) copolymer with DMAA (CPMA/DMAA), already possessing the needed aldehyde groups and therefore assumed not to need LO to sufficiently crosslink gelatin, was also designed and synthesized. CPMA/DMAA was prepared to investigate the influence of the oxidative activity of LO on the gelatine reticulation process, regardless of the presence of already oxidized aldehyde functions. Oxidation tests of amine-containing copolymers with LO were performed to identify the best LO substrates to be used in subsequent experiments of gelatine reticulation. Copolymers that demonstrated to be excellent substrate for LO (CP5/DMAA, CP11b/DMAA, and CP11c/DMAA) were employed with CPMA/DMAA in crosslinking tests with gelatine or gelatine/HA systems. The percentage of crosslinking was determined in all samples by UV and acid/base titrations to detect the sample(s) suitable for further investigations, resulting in sample M21 obtaining crosslinking gelatine with CP/DMAA as the best candidate. Since gelatine is surely biocompatible [49], the possible biocompatibility of M21 was then evaluated by considering the results obtained from cytotoxicity studies previously carried out with the copolymer P5 structural analogous of CP5/DMAA [50].

## 2. Results and Discussion

Our previous studies regarded copper-containing amino oxidases (CAO), including the abovementioned LO, a class of enzymes involved in important cellular processes such as the collagen crosslinking shown in Figure 1, cell proliferation, and the removal of biogenic amines by oxidation. Such studies led to the preparation of a linear copolymer containing amino-butyl units (P1c) (Appendix A) [51] that proved to be an excellent substrate of LO, 9.7 times more effective than the natural LO substrate elastin, and therefore of interest for this new work [51]. Additionally, in a more recent work, Pc1 (renamed P5) demonstrated remarkable broad-spectrum bactericidal effects against several multidrug-resistant clinical isolates [52]. Such property could be of paramount importance in the future application of a P5/gelatine scaffold to treat chronic wounds, like those occurring in diabetes, susceptible to bacterial infections. The presence of gelatine could offer an ECM-like environment for cell adhesion and proliferation, thus promoting wound healing [53], while that of the bactericidal P5 could prevent detrimental infections [54]. It was then thought that also variants of P1c, combined with the oxidative action of LO, could have acted as crosslinkers of gelatine, as shown in Figure 2 and Figure 3, and could provide the desired material with mechanical properties superior to those of pristine gelatine.

We started our study by synthesizing the monomer **5** (Appendix A) needed to prepare copolymers structural analogs of P5 shown in Appendix A and performing copolymerization reactions of **5** with DMAA according to previously reported procedures [51,52]. Subsequently, a new terpolymer containing DMAA, acrylic acid (AA), and **5** was also prepared. Three amidoamine type monomers (**11a**, **11b**, and **11c**) (Appendix A), more biodegradable than **5** due to the presence of acrylic functions in place of the styrene one, were prepared containing C2, C4, and C6 alkyl chains, as linkers between the amido and the amino groups. Such monomers were polymerized with DMAA, achieving the corresponding copolymers CP11a/DMAA, CP11b/DMAA, and CP11c/DMAA.

### 2.1. Synthesis and Characterization of Styrene-Based Monomer ***5*** and Its Copolymers

Figure 4 shows the previously reported synthetic procedure necessary to prepare monomer **5** and its copolymers [51,52]. In this work, CP5/DMAA copolymers were prepared with different concentrations of **5** according to data reported in Appendix A. 

All the intermediates and compound **5** were characterized by FTIR, ^1^H, and ^13^C NMR analyses, GC-MS chromatography, or elemental analyses, and when possible, melting points were determined. All results agreed with those previously reported [51,52].

Copolymers were fractioned as reported [52] and characterized first by FTIR spectroscopy, obtaining results matching those in literature [52] and then by ^1^H NMR analysis, evidencing a peak at 2.90 ppm attributable to the CH_3_ groups of DMAA.

For copolymer CP5/DMAA-42.9 hydrochloride (HCl), taken as reference compound, the NH_2_ equivalents [42], the average molecular mass (Mn) [42], the average hydrodynamic diameter of particles (Z _AVE_), as well as the related poly-dispersion index (PDI) and Z-potential (ζ-p) were assessed as previously reported [51,52,55], while potentiometric titrations were performed as more recently described [56]. Results from all the above-mentioned characterizations agreed with those previously described for P5 and have been reported in Appendix A.

A new three-component copolymer containing DMAA, AA, and **5** (terpolymer, TP5/DMAA/AA) was prepared to have a polymeric material in which the hydrophilicity and the ability to establish secondary interactions could be varied in a simple way, for example, by changing the pH.

TP5/DMAA/AA was obtained starting from a molar ratio of monomers in the feed of **5**/DMAA/AA = 1/8/1 (Figure 5).

After isolation, the terpolymer was purified by dissolution/precipitation in MeOH/Et_2_O, obtaining a material free from traces of monomers as demonstrated by the ^1^H NMR spectrum (300 MHz, CDCl_3_), which did not present the characteristic signals of the vinyl systems. The FTIR spectrum showed the presence of a very intense band at 1637 cm^−1^ relating to the amide carbonyl of DMAA and a much less intense band at 1720 cm^−1^ attributable to the carboxyl function of AA.

Solubility tests were carried out in different solvents on the synthesized compounds and on the DMAA homopolymer (HP-DMAA) prepared for comparison purposes. Representative results have been included in Appendix A.

### 2.2. Synthesis and Characterization of Amidoamine Monomers ***11a–c*** and Their Copolymers with DMAA

Monomers **11a–c** (Appendix A) were prepared by to perform a three-step procedure reported in the literature [57], starting from the corresponding commercially available diamines.

#### 2.2.1. Synthesis of *N*-Boc-Protected α-ω-di-Aminoalkanes **6a–c**

Since the monoacylation of α-ω-di-aminoalkanes is unfavorable [57], to avoid the formation of *bis*-acrylamides, endowed with a strong tendency to polymerize and crosslinking, it was preferred to protect one of the two amine functions with the *t*-butoxy carbonyl group (Boc). Unfortunately, even upon this protection, small percentages of double acylation products formed. Anyway, since these side products were relatively stable and possessed a solubility very different from that of the products of monoacylation, they were easily separated from the desired Boc-protected compounds, which were effortlessly purified. Specifically, the monoprotected diamines **6a** and **6b** were synthesized according to Figure 6.

In the FTIR spectra of both **6a** and **6b,** strong bands belonging to the C=O stretching of the Boc group were observable at 1695 and 1705 cm^−1^, respectively, while bands at 3663 and 3358 cm^−1^, were due to the NH stretching of amine and amide groups (Appendix A). Bands due to the CH stretching of methyl and methylene groups were observable at 2977, 2933, 2870 cm^−1^ and at 2976, 2932, and 2865 cm^−1^, respectively. The ^1^H NMR spectrum of **6a** showed a signal integrable for two protons due to the NH_2_ group at 1.28 ppm, an intense signal (9H) at 1.44 ppm related to the methyl groups of the *t*-butyl group, while a triplet at 2.79 ppm and a quartet at 3.16 ppm related to the methylene groups linked to the urethane NH and to the primary amine group respectively (Appendix A). The broad singlet at 5.49 ppm was due to the amide NH. The ^13^C NMR spectrum of **6a** presented the intense signal of the CH_3_ of *t*-butyl group at 28.43 ppm, the signals of methylene groups of the aliphatic chain at 41.88 and 43.41 ppm, while the weak signals at 79.21 ppm and 156.27 ppm were due to the quaternary carbons of the *t*-butyl group and of carbonyl respectively (Appendix A).

Although the ^1^H NMR spectrum of **6b** was very similar to that one of **6a** (a singlet at 1.17 ppm, a very intense signal that can be integrated for nine protons at 1.44 ppm, a triplet at 2.71 ppm, a quartet at 3.11 ppm and a broad singlet at 5.21 ppm), an additional multiplet in the range 1.46–1.54 ppm relating to the two methylene groups inside the aliphatic chain was detectable (Appendix A).

The FTIR spectrum of the product of double acylation product **7a** formed during the synthesis of **6a** showed a very intense band at 1684 cm^−1^ relating to the two urethane carbonyls and an intense and sharp band at 3382 cm^−1^ due to the NH stretching.

The ^1^H NMR spectrum was very simple due to the symmetry of the molecule and presented only three signals. A very intense peak that can be integrated for eighteen protons due to the methylene groups of *t*-butyl was detectable at 1.44 ppm, followed by the singlet of the methylene groups at 3.22 ppm and by the broad singlet of the NH groups at 5.19 ppm. Similarly, also the ^13^C NMR spectrum was extremely simple, with only four signals. A very intense peak was detectable at 28.42 ppm (methyl groups), followed by that of the two methylene groups (40.82 ppm), by that of the quaternary carbons of *t*-butyl (79.33 ppm), and finally by the signal of the carbonyl groups (156.44 ppm).

In the case of compound **6c**, the protection of the amino group with the Boc group was achieved more efficiently with the reagent *S*-*t*-butoxycarbonyl-4,6-dimethyl-2-mercaptopyrimidine (**8**) [58,59] according to the Figure 7. 

A large excess of commercially available 1,6-diaminohexane (3.3 equiv.) was used to minimize the formation of the product of double protection (**7c**), and the desired monoprotected diamine **6c** was isolated and characterized both as hydrochloride and as free base Particularly, in the FTIR spectrum of the hydrochloride salt of **6c** (**6c HCl**), the urethane carbonyl band was identified at 1687 cm^−1^, while that of the NH_3_^+^ group was detected at 3378 cm^−1^. Bands due to the CH stretching of methyl and methylene groups were observable at 2983, 2936, and 2866 cm^−1^ (Appendix A). Otherwise, the FTIR spectrum of **6c** showed bands at 1695 cm^−1^ (C=O), 3344 cm^−1^ (NH and NH_2_), and at 2976, 2930, and 2857 cm^−1^ (CH stretching of methyl and methylene groups) (Appendix A). In the ^1^H NMR spectrum of **6c HCl**, we identified two multiplets integrated, each one for four protons and centered around 1.37 and 1.58 ppm relating to the four methylene groups inside the aliphatic chain and a very intense singlet at 1.43 ppm belonging to the methyl groups of Boc. The methylene groups bonded to the NH_3_^+^ and urethane nitrogen atom gave two partially overlapping triplets between 2.97 and 3.08 ppm (Appendix A). The ^13^C NMR spectrum of **6c HCl** showed nine signals. Six signals at 25.39, 25.54, 26.79, 28.87, 39.66, and 40.06 ppm as regards the methylene chain, two signals at 27.97 and 80.89 ppm concerning the methyl groups and quaternary carbon atom of the *t*-butyl system, and one signal at 158.42 ppm regarding the carbonyl group (Appendix A). The free amine **6c** provided a ^1^H NMR spectrum where, in the range of 1.25–1.55 ppm, we found a multiplet signal due to the four methylene groups inside the aliphatic chain. The signal of the two protons of the amino group and the very intense singlet signal of the methyl groups of the *t*-butyl system were superimposed and detectable at 1.44 ppm. The methylene groups bonded to the urethane and amino nitrogen atoms gave a triplet at 2.68 ppm and a multiplet at 3.10 ppm, respectively, while the amide NH provided a broad singlet at 4.69 ppm (Appendix A). Like that of **6c HCl**, the ^13^C NMR spectrum of free amine **6c** showed nine signals at 26.55, 26.64, 30.09, 33.61, 40.53, 42.05, 28.45, 78.97 and 156.06 ppm (Appendix A). As for **7c**, the FTIR spectrum showed mainly the band of the two urethane carbonyls at 1687 cm^−1^, while that of the NH groups was detected at 3442 cm^−1^. The ^1^H NMR spectrum of **7c** showed a multiplet integrable for four protons relating to two of the four methylene groups inside the aliphatic chain at 1.32 ppm. A very intense singlet due to the six methyl groups of the *t*-butyl systems, which overlapped almost completely with the signal of the other two methylene groups, was detected at 1.44 ppm. At 3.10 ppm, there was a poorly resolved quartet integrable due to four protons relating to methylene groups linked to the nitrogen atoms, whose protons gave a broad singlet at 4.66 ppm. The ^13^C NMR spectrum of this molecule showed six signals. A very intense singlet was observable at 28.45 (six methyl groups), three signals were found at 26.37, 30.03, and 40.43 ppm (six methylene groups), one signal was detected at 78.99 ppm (two quaternary sp3 carbons), and finally, one small peak was at 156.07 ppm (two carbonyls).

#### 2.2.2. Synthesis of Acryloyl Monomers **11a–c**

Monomers **11a**–**c** were obtained from the Boc-protected diamines **6a**–**c** according to Figure 8).

Characterization of Intermediates **10a–c**.

In the FTIR spectra of compounds **10a**–**c**, the band of the urethane carbonyl was detected at 1696 (**10a**), 1687 (**10b**) and 1679 cm^−1^ (**10c**), while that of carbonyl conjugated with the vinyl system of amide was observable at lower wavenumbers, such as 1662 (**10a**), 1661 (**10b**) and 1652 cm^−1^ (**10c**). The double bond of the vinyl group gave bands at 3078 cm^−1^ (**10a**) and 3040 cm^−1^ (**10b**) due to the =C–H stretching (not detectable for **10c**) and bands in the range 957–995 cm^−1^ due to the =C–H banding for all compounds. Bands due to the NH groups were found at 3304 cm^−1^ (**10a**), at 3309 and 3345 cm^−1^ (**10b**) and at 3433 cm^−1^ (**10c**) (Appendix A).

In the ^1^H NMR spectrum of **10a**, the singlet of the proton atoms of the *t*-butyl group was found at 1.43 ppm, the multiplets of the aliphatic methylene groups at 3.31 and 3.44 ppm, while the vinyl system gave three doublets at 5.63, 6.13 and 6.26 ppm. The proton atoms bound to the amide and urethane nitrogen atoms gave two broad signals at 5.19 and 6.74 ppm (Appendix A).

The ^13^C NMR spectrum of **10a** showed an intense signal of the three CH_3_ of the Boc group at 28.38 ppm, the signals of the methylene groups linked to the nitrogen atoms at 40.15 and 40.91 ppm, that of the quaternary carbon of the *t*-butyl system at 79.76 ppm, while the signals of the vinyl carbon atoms were observable at 126.23 and 130.96 ppm. Lastly, the signals of the two carbonyl groups were found at 157.13 and 166.32 ppm (Appendix A).

The ^1^H NMR spectrum of **10b** showed signals at 1.44 ppm (singlet of the *t*-butyl group), in the range 1.51–1.60 ppm (multiplet of the methylene groups inside the aliphatic chain), at 3.12 ppm (multiplet) and at 3.33 ppm (quartet) (methylene groups bonded to nitrogen atoms), at 5.61, 6.16 and 6.27 ppm (three doublets, vinyl system), as well as at 4.81 and 6.55 ppm (broad singlets due to the proton atoms bound to the amide and urethane nitrogen ones) (Appendix A).

The ^13^C NMR spectrum of **10b** revealed ten signals. Specifically, two signals were due to the carbon atoms of the methylene groups inside the aliphatic chain (26.55 and 27.20 ppm), one signal was due to the methyl groups of Boc (28.44 ppm), two owed to the methylene groups linked to the nitrogen atoms (39.23 and 40.14 ppm), one belonged to the quaternary carbon atom of the *t*-butyl system (79.24 ppm), two belonged to the vinyl group (126.03 and 131.09 ppm) and the two were due to the carbonyl carbon atoms (156.27 and 165.84 ppm) (Appendix A).

In the ^1^H NMR spectrum of **10c**, the intense singlet of the *t*-butyl protons (1.44 ppm) overlapped with the multiplet signal of the four protons belonging to the two more internal methylene groups of the aliphatic chain (1.44 and 1.54 ppm). The signal of the other two methylene groups inside the chain provided a multiplet at 1.34 ppm. At 3.09 and 3.30 ppm, the quartets of methylene groups bonded to nitrogen atoms were identified, while the vinyl system presented three doublets at 5.60, 6.19, and 6.27 ppm. The protons bound to the amide and urethane nitrogen atoms gave two broad signals at 4.82 and 6.67 ppm (Appendix A). 

Overwise, in the ^13^C NMR spectrum of **10c**, twelve signals were detected. Specifically, we observed the signals of the aliphatic carbons internal to the chain (25.89, 26.26, 29.33 and 29.98 ppm), that of the methyl groups of the *t*-butyl system (28.44 ppm), those of carbons bonded to nitrogen atoms (39.23 and 40.20 ppm), the signal of the quaternary carbon of Boc (79.05 ppm), the signals of the vinyl carbon atoms (125.84 and 131.22 ppm) and the signals of the two carbonyl carbons (156.25 and 165.84 ppm) (Appendix A).

Characterization of Monomers **11a–c**.

The FTIR spectra of monomers **11a**, **11b**, and **11c** showed bands belonging to the vinyl system in the range 946–999 cm^−1^ (=C–H bending), as well as at 3066 cm^−1^ (**11a**) and 3073 cm^−1^ (**11b**) (=C–H stretching). The carbonyl groups provided bands at 1656 cm^−1^ (**11a**), 1651 cm^−1^ (**11b**) and 1653 cm^−1^ (**11c**), while the intense absorption of NH_3_^+^ groups was found at 3243 cm^−1^ (**11a**), 3262 cm^−1^ (**11b**), as well as at 3411 and 3306 cm^−1^ (**11c**) (Appendix A).

Since the ^1^H NMR spectra of all monomers **11a**–**c** were acquired in CD_3_OD, H_2_O, or a mixture of two solvents, the proton atoms linked to the nitrogen atoms were not detectable because they were exchangeable. In all spectra, two triplets were detected at 3.09 and 3.53 ppm (**11a**), 3.03 and 3.32 ppm (**11b**), and at 3.01 and 3.28 ppm (**11c**) relating to the methylene groups linked to the amide and NH_3_^+^ functions. Similarly, the vinyl systems of all compounds gave a quartet (**11a**) or a doublet (**11b** and **11c**) integrable for a proton centered around 5.72 ppm (**11a**), 5.75 (**11b**) and 5.76 (**11c**), as well as a multiplet integrable for two protons positioned around 6.25 ppm (**11a** and **11b**) and 6.24 (**11c**) (Figure 2, Figure 3 and Figure 4). More complex respect that of **11a** were the spectra of **11b** and **11c,** which showed an additional multiplet in the range of 1.58–1.78 ppm relating to the methylene groups inside the aliphatic chain (**11b**) (Figure 3) and two multiplets in the range 1.39–1.63 ppm relating to the four internal methylene groups of the aliphatic chain (**11c**) (Figure 4).

The ^13^C NMR spectra of **11a**, **11b**, and **11c** presented five, seven, and nine signals, respectively. All showed the two methylene carbons bound to the nitrogen atoms at 38.85 and 41.48 ppm (**11a**), 37.92 and 38.56 ppm (**11b**), as well as at 39.22 and 39.45 ppm (**11c**). Similarly, all spectra showed two signals belonging to the vinyl systems at 128.13 and 132.16 ppm (**11a**), 126.21 and 129.60 ppm (**11b**) as well as at 126.99 and 130.11 ppm (**11c**) and the signal of the carbonyl group at 169.90 ppm (**11a**), 167.45 ppm (**11b**) and 168.49 ppm (**11c**) (Figure 5, Figure 6 and Figure 7). Also, the spectrum of **11b** showed two additional signals due to the two methylene groups inside the aliphatic chain at 23.80, 25.13 (Figure 6), while the spectrum of **11c** presented four additional signals at 25.19, 25.47, 26.61, 28.02, due to the four methylene groups inside the aliphatic chain (Figure 7).

### 2.3. Preparation of Copolymers of ***11a–c*** with DMAA (CP11a–c/DMAA)

Monomer **11c**, which does not demonstrate a tendency to self-polymerize, was easily converted into the desired polymeric structures according to Figure 9.

The copolymers were purified through three cycles of dissolution/precipitation by MeOH/Et_2_O to eliminate any traces of the monomers. A copolymer sample was also fractionated at 25 °C, using the couple solvent/non-solvent MeOH/Et_2_O. The FTIR spectrum of the CP11c/DMMA showed the amide carbonyl band at 1629 cm^−1^ and the amine group band at 3342 cm^−1^. In the ^1^H NMR spectrum, the signals of the vinyl system at 5.76 and 6.15–6.32 ppm of the monomer **11c** disappeared, while the signal of the methyl groups of DMAA units was observable at 2.95 ppm. Otherwise, to obtain the polymeric structures containing units of **11a** and **11b**, the Boc-protected monomers **10a** and **10b** were copolymerized in place of **11a** and **11b** according to Figure 10. The obtained copolymers were subsequently subjected to a deprotection reaction in acidic conditions (Figure 10).

After an initial purification by dissolution/precipitation in MeOH/Et_2_O, CP10a/DMAA and CP10b/DMAA were also fractionated at 25 °C, obtaining polymeric materials free of monomers. The ^1^H-NMR spectra of both copolymers were characterized by two narrow signals at 1.43 ppm due to the *t*-butyl systems and by a signal at 2.91 ppm due to the methyl groups linked to the amide nitrogen atoms of the DMAA. The ^1^H NMR spectra were used to estimate the content of monomers **10a** and **10b** in the prepared copolymers. Particularly, from the calculation of the ratio between the integral of the signals at 1.43 ppm of the *t*-butyl group and that of the signals at 2.91 ppm (methyl groups of DMAA), percentage compositions in the range of 8.8–9.9% were obtained for CP10a/DMAA and of 6.8–8.1% for CP10b/DMAA. These values were very close to the percentage composition of the monomers in the copolymerization feed. CP10a/DMAA and CP10b/DMAA were deprotected by adding a solution of 3N HCl in EtOAc to the solid copolymers, thus achieving CP11a/DMAA and CP11b/DMAA as sticky solids. To fully remove the solvent and water, it was necessary to treat the polymeric materials with hot EtOH, followed by evaporation twice and repeating the treatment with toluene. At the end of these operations, the copolymers appear as white glassy solids. The occurred deprotection was demonstrated by the FTIR spectra, which lack the urethane band around 1680–1700 cm^−1^, and by the ^1^H NMR spectra, where the intense signal of the *t*-butyl group around 1.43 ppm was no longer visible.

The solubility of all prepared copolymers was assayed in different solvents, and results have been reported in Appendix A.

### 2.4. Preparation of Methacrolein (MA)/DMAA Copolymer (CPMA/DMAA)

In addition to the amine-containing copolymers described in the previous sections, designed to be possible substrates of LO, a copolymer already containing the aldehyde functionality was prepared. It would have allowed us to explore different crosslinking conditions and to make comparisons between the crosslinking behavior of gelatine in the presence of enzymatically in situ oxidized copolymers and copolymers already containing aldehyde functionalities. To this end, commercially available methacrolein (MA) was copolymerized with DMAA according to Figure 11.

The copolymer was purified by dissolution and precipitation in MeOH/Et_2_O, obtaining a material free from traces of monomer, as was highlighted by examining the ^1^H NMR spectrum, where the signals of the vinyl system are no longer visible. The FTIR spectrum evidenced the presence of the two carbonyls, showing the intense band of amide one at 1630 cm^−1^ and the less intense band of the aldehydic one at 1720 cm^−1^. The ^1^H NMR spectrum demonstrated the signal of the methyl groups of the DMAA units at 2.90 ppm and that of the aldehyde protons in the range 9.00–9.60 ppm as an enlarged and complex signal, probably due to the presence of different types of aldehyde groups in the copolymer backbone. The ^1^H NMR spectrum was used to estimate the MA content in the copolymer. From the calculation of the ratio between the integral of the signal at 2.90 ppm and that of the signals at 9.00–9.60 ppm, a percentage composition of 22.7% of MA was obtained. This value was slightly higher than the percentage composition of the monomer in the copolymerization feed. The solubility of the CPMA/DMAA copolymer was found to be very similar to that of the DMAA homopolymer, being insoluble in petroleum ether 40–60 °C and ethyl ether, soluble in dioxane, acetone, CHCl_3_, DCM, MeOH, DMF, DMSO, H_2_O, and swelling in hot THF and toluene.

### 2.5. Experiments of Gelatine Crosslinking

#### 2.5.1. Gelatine

The main characteristics of gelatine (Gel B) used in this study are summarized in Appendix A. The determination of the amino groups in Gel B was performed both via acid-base titration following a literature procedure applied to Gel B-type gelatines [60] and spectrophotometrically.

By acid-base titration curve, applying Cannan’s criteria [61], it was possible to obtain the minimum, maximum, and average values of the lysine groups (NH_2_ content) in the pH range of 8.5–11.5 In our case, they were found to be 0.189 mmol/g (pH = 8.5), 0.593 mmol/g (pH = 11.5), proving the average value of 0.391 mmol/g reported in Appendix A. According to the literature, a value close to 0.32 mmol/g was determined by other authors with the same procedure for gelatine of a similar type [62].

For the spectrophotometric determination of the content of NH_2_ groups, it was followed a procedure based on the reaction of the NH_2_ groups of gelatine with 2,4,6-trinitrobenzenesulfonic acid (TNBS) in a basic environment [63]. The subsequent acidic hydrolysis breaking down the peptide bonds of the gelatine, caused the release of the labeled lysine moieties, thus providing the tri-nitrophenyl chromophore according to Appendix A, which was quantified by UV analyses at 346 nm.

As observable in Appendix A, the value found in this case (0.219 mmol/g) was lower than that obtained by titration with NaOH. This may depend on an incomplete functionalization of Gel B. However, the spectrophotometric method remains very attractive due to the minimal quantities of sample needed for the test (approximately 11 mg) when compared with those for the titration with NaOH (220–250 mg).

#### 2.5.2. Enzymes

For the enzyme interaction and crosslinking assays, we used LO isolated by us from the bovine aorta [64] by applying the first part of the protocol found in the literature [65]. The main steps of the extraction process have been reported in Appendix A. The SDS-PAGE electrophoretic analysis of the preparation isolated by us evidenced the presence of LO in correspondence with the 32 KDa band. The activity of the enzyme was estimated according to a fluorometric method [64,66] against a calibration curve constructed using H_2_O_2_ solutions of known concentrations in the presence of homo vanillic acid and peroxidase enzyme. It is, in fact, known that H_2_O_2_, in the presence of peroxidase, oxidizes homo vanillic acid to a dimeric fluorescent species quantifiable by fluorimetry [67] (Appendix A). By subsequently measuring the fluorescence of solutions containing homo-vanillic acid, peroxidase, LO, and 1,4-diaminobutane as a LO substrate molecule, it was possible to measure the H_2_O_2_ released in the diamine oxidation reaction and estimate the enzymatic activity of our preparation, which resulted around 0.81 nmol/mg of wet weight.

For comparison purposes, the bovine enzyme plasma amine oxidase (PAO) with an activity of 22.5 units/mg of dry weight was procured from Biochemical Corporation (Worthington, NJ, USA) and used in our experiments.

#### 2.5.3. Enzymatic Oxidation Assays

To quickly select the most suitable polymeric materials to be used in the reticulation experiments with gelatine, their susceptibility to oxidation with LO and PAO was verified using the Schiff reagent. Specifically, the Schiff reagent highlights the presence of aldehyde groups through the development of a typical fuchsia coloration. Appendix A summarizes the results obtained from the experiences carried out.

The test was also performed on the homopolymer of *N*, *N*-di-methylacrylamide (HP-DMAA) used as blank to check for possible matrix interferences, with negative results. Among the 4-aminobutylstyrene copolymers investigated, those with a higher content of **5** (CP5/DMAA-43.1) (Exp 2–3) gave a clear positive response to the Schiff test with both LO and PAO. Those with a lower content of **5** (Exp 4–7) provided weak staining when treated with LO and no staining with PAO. Positive responses to the Schiff test were recorded both with LO and with PAO, with the copolymers CP11b/DMAA-10.0 and CP11c/DMAA-9.9, characterized by a sequence of 4 and 6 methylene groups, respectively, between the amine nitrogen atom and amide one.

#### 2.5.4. Dynamic Light Scattering (DLS) Analyses of Selected Copolymers

Copolymers that gave positive results in the Schiff test and CPMA/DMAA were selected for experiments of gelatine crosslinking and were therefore analyzed to determine their particle size (Z _AVE_, nm), polydispersity (PDI), and surface charge before crosslinking reactions. Specifically, the samples analyzed were CP11b/DMAA (*n* = 4), CP11c/DMAA (*n* = 6) and CPMA/DMAA. The results have been reported both in Appendix A and in Figure 8. The data related to the previously determined copolymer CP5/DMAA were also included in Appendix A.

Except for CP11b/DMAA copolymer, which demonstrated micro-sized particles (2.6 µm), other copolymers were nanosized with particle dimensions in the range 112–373 nm, being the copolymer prepared with MA one demonstrating the smallest dimensions (112 nm) and the copolymer obtained using the amidoamine monomer with the C6 linker that one showing the largest dimensions (373 nm). Amidoamine-type copolymers, although very different in particle size, demonstrated similar low PDI, while the styrene copolymer obtained by monomer **5** showed PDI > 1. As expected, all ammonium-containing copolymers displayed positive ζ-p, CP5/DMAA, having the highest value. Anyway, also, the MA-derived copolymer not possessing cationic groups provided a positive value of ζ-p, even higher than that given by CP11b/DMAA (+18.3 vs. +6.5). Although polymer particles with high positive ζ-p could act as membrane disruptors, thus being cytotoxic on normal cells, it was reported that cellulose scaffolds modified with positive moieties, thus demonstrating ζ-p of +15–25 mV, increased cell attachment by 70% compared to unmodified cellulose [68].

#### 2.5.5. Gelatine Crosslinking Experiments

Fourteen Gel B crosslinking experiments, which differed for the percentage (wt%) of copolymer, buffers, or enzymes used, were performed, and seven representative examples have been included in Appendix A.

#### 2.5.6. Titrations of Crosslinked Gelatines

Chemical crosslinking methods, such as the one proposed in this study, which use the amino groups of the lysine or hydroxylysine residues of gelatine for reticulation, leave on crosslinked material non-reacted NH_2_ groups. Their amount can be determined with spectroscopic methods or by acid-base titration as previously described for pristine gelatine. Once the residual NH_2_ groups are quantified, the result can be used to calculate the crosslinking percentage if the NH_2_ content in the starting material is known. Following the same procedure of functionalization of the gelatine previously described, some selected crosslinked samples, as reported in Appendix A, were reacted with TNBS, and the unreacted NH_2_ was quantified spectrophotometrically. Results are available in Appendix A.

For sample M21, a very good crosslinking value was recorded by UV titration; for M31, the crosslinking was very modest both according to UV and acid-base titrations, and the compound was not further considered. For samples M23 and M26 obtained by crosslinking Gel B in the absence of enzyme and with the MA/DMAA copolymer CPMA/DMAAA-20.0, the UV titration provided absorbance values very close to those of non-crosslinked gelatin with a percentage of crosslinking close to zero. Anyway, it should be considered that in these samples, the most probable crosslinking mechanism was via imine-type bonds, as previously discussed, which could be unstable to the hot HCl treatment necessary in the UV titration. Consequently, it could be assumed that the crosslinking bonds have been destroyed during UV analysis, thus causing the decrease of absorbance values close to those of non-crosslinked Gel B. For these samples, titration with NaOH was therefore carried out in analogy to what was conducted previously for Gel B to have more reliable information. The results of the titration, reported in Appendix A, confirmed that crosslinking also occurred for these samples.

The M32 sample obtained by crosslinking Gel B with the same CPMA/DMAAA-20.0 but in the presence of hyaluronic acid (HA) provided an appreciable crosslinking value (20.5%) also by UV titration, indicating a greater resistance to hydrolysis with HCl of the hybrid material Gel B/HA. The crosslinking determined for M32 by titration with NaOH was, as expected, decidedly higher (Appendix A).

It is interesting to note that the M34 sample obtained by crosslinking Gel B with the CPMA/DMAAA-20.0 copolymer in the presence of LO showed a decidedly higher crosslinking percentage (46.1%) compared to that of the M23 sample (5.5%) suggesting the importance of enzyme in the formation of crosslinked structures of gelatin via non-hydrolyzable bonds.

#### 2.5.7. Cytotoxicity Experiment on P5 Structural Analogous of CP5/DMAA

From experiments discussed in the previous Section 2.5.6, it was evidenced that the most promising crosslinked materials for further studies were those obtained using copolymers of CPMA/DMAA type and of CP5/DMAA type due to their reticulation percentage. Among these, the sample obtained utilizing CP5/DMAA (M21) demonstrated the highest crosslinking percentage (70.5%) also by UV determinations, matching that reported for a gelatine crosslinked sample and detected with the same method [69]. When tested on human chondrocytes, such samples showed good adhesion, viability, and proliferation, as well as extensive 3D scaffold colonization [69]. In this regard, since the biocompatibility of gelatine is extensively reported [49], before performing further physicochemical and biological investigations, the possible biocompatibility of the synthetic constituent of M21 (CP5/DMAA) was evaluated by considering the results obtained from cytotoxicity studies previously reported on the copolymer P5, structural analogous of CP5/DMAA [50]. Figure 9 and Figure 10 show the results from dose and time-dependent cytotoxicity studies carried out with P5 on two human neuroblastoma cell lines (IMR-32 and SH SY5Y).

Both cell lines were treated with P5 at a concentration of 0.5–76.5 µg/mL (0.1–15 µM) for 24, 48, and 72h. The viability of IMR-32 cells was not significantly reduced when exposed to P5 for 24 and 48 h at all concentrations tested, while a slight reduction in cell viability was observed only for high times of exposure (72 h) and at concentrations > 51.0 µg/mL (10 µM). Even less cytotoxic effects were observed when SH SY5Y cells were treated with P5. In this case, the viability of SH SY5Y cells was not significantly reduced for all times and at all concentrations tested, while a significant proliferation was observed.

## 3. Materials and Methods

### 3.1. Chemicals and Instruments

All reagents and solvents were from Merck (formerly Sigma-Aldrich, Darmstadt, Germany). The solvents were dried and distilled according to standard procedures. 2,2’-azobis-(2-methylpropionitrile) (AIBN) was crystallized from methanol. 4-chlorostirene was distilled at reduced pressure under nitrogen and stored at −20 °C. *N*, *N*’-dimethyl-acrylamide (DMAA) was distilled under reduced pressure and nitrogen and stored at −20 °C. The organic solutions were dried over anhydrous magnesium sulfate and were evaporated using a rotatory evaporator operating at a reduced pressure of about 10–20 mmHg. The melting ranges of the solid compounds in this study were determined on a 360 D melting point device with a resolution of 0.1 °C (MICROTECH S.R.L., Pozzuoli, Naples, Italy). Melting points and boiling points are uncorrected. The Fourier transform infrared (FTIR) spectra were recorded on a Perkin Elmer System 2000 FTIR spectrophotometer (Perkin Elmer, Milan, Italy) as liquid films or KBr tablets, while attenuated total reflectance (ATR) FTIR was carried out on the same instrument as previously reported [70]. Dynamic Light Scattering (DLS) and Z-potential (ζ-p) determinations were performed using a Malvern Nano ZS90 light-scattering apparatus (Malvern Instruments Ltd., Worcestershire, UK). Column chromatography was performed on Merck silica gel (70–230 mesh) (Merk, Milan, Italy). Thin layer chromatography (TLC) was carried out using aluminum-backed silica gel plates (Merck DC-Alufolien Kieselgel 60 F254, Merck, Washington, DC, USA), and detection of spots was made by UV light (254 nm) using a Handheld UV Lamp, LW/SW, 6W, UVGL-58 (Science Company^®^, Lakewood, CO, USA). Mass spectra were performed on a Varian Saturn 2000 GC-MS ion trap instrument (Varian, Inc., Palo Alto, CA, USA) equipped with a DB-5MS column (30 m, i.d. 0.25 mm) from J&W Scientific (Agilent, Milan, Italy). The HPLC analyses were conducted on a JASCO LC-980 system (JASCO EUROPE S.R.L., Lecce, Italy) complete with UV detector, degasser, gradient, oven, and equipped with a Hypersil ODS 5 µ column (25 × 0.46 cm) (JASCO EUROPE S.R.L., Lecce, Italy), using CH_3_CN/H_2_O 6/4 as eluent mixture. The ^1^H-NMR and ^13^C-NMR spectra were carried out on a Bruker DPX 300 (Bruker Italia, s.r.l., Milan, Italy) spectrometer using TMS as an internal reference. The multiplicity of peaks was expressed using s (singlet), d (doublet), dd (double doublet), t (triplet), q (quartet), m (multiplet) and br s (broad signal). The UV spectra were recorded on an HP 8453 UV-visible System spectrophotometer (Agilent, Milan, Italy) using quartz cuvettes with an optical path of 1 cm. Fluorescence measurements were carried out using a Cary Eclipse Fluorescence Spectrometer (Agilent, Milan, Italy). Potentiometric titrations were performed with a Hanna Micro-processor Bench pH Meter (Hanna Instruments Italia srl, Ronchi di Villafranca Padovana, Padova, Italy). The molecular weight of the CP9/DMAA-42.9 copolymer was determined on a Knauer K-7000 osmometer (Knauer, Berlin, Germany) in MeOH at 45 °C. Dialyses were performed using a CelluSep T3 cylindrical membrane (MWCO 12,000–14,000) (Euroclone S.p.a., Milan, Italy). Bovine plasma amine oxidase (PAO, 22.5 unit/mg DW) was purchased from Biochemical Corporation, Worthington, NJ, USA. The gelatine used in this study was type B gelatine (Sigma-Aldrich, Darmstadt, Germany), referred to as Gel B.

### 3.2. Synthesis and Characterization of Styrene-Based Monomer ***5*** and Its Copolymers

The synthesis of monomer 4-ammino-butyl-styrene hydrochloride (**5**) and of its copolymers obtained using different concentrations of **5** (CP5/DMAA-x), where x indicates the percentage of **5** in the polymerization feed, as well as the procedure used for fractioning the obtained copolymers, have been described in our previous papers [51,52]. The experimental data of all copolymerization reactions carried out in this study are available in Appendix A.

#### Preparation of **5**/DMAA/Acrylic Acid (AA) Terpolymer (TP5/DMAA/AA)

In a 10 mL tailed test tube, equipped with a magnetic stirrer and a screw cap with silicone septum, **5** (103.8 mg, 0.49 mmol) and AIBN (1 mg, 0.006 mmol) dissolved in 1.5 mL of anhydrous *N*, *N*’-dimethylformamide (DMF) were introduced under nitrogen atmosphere. Upon solubilization, the mixture was transferred via a steel needle into a 10 mL three-necked flask having, in turn, a screw cap with silicone septum. After washing the needle with anhydrous DMF (0.5 mL), DMAA (388.6 mg, 3.92 mmol, 400 μL) and AA (35.3 mg, 0.49 mmol, 34 µL) were introduced via a micro-syringe. The polymerization mixture was degassed by bubbling nitrogen for 15 min and then kept under stirring at 60 °C in an oil bath for 20 h. Subsequently, the mixture was diluted with anhydrous DMF (2 mL) and precipitated in 100 mL ethyl ether (Et_2_O). The copolymer in the form of a white powdery solid was brought to constant weight at reduced pressure (258.7 mg, 49.0% yield).

FTIR (KBr, cm^−1^): 3427 (NH), 1721 (COOH), 1637 (C=O). ^1^H NMR (DMSO-*d6*, 300 MHz, δ ppm): 2.79 (CH_3_ DMAA), 8.00 (OH AA).

### 3.3. Synthesis and Characterization of Acryloyl Monomers (***11a–c***) and Their Copolymers (CP11a–c/DMAA)

#### 3.3.1. Synthesis and Characterization of *N*-*t*-Butoxy Carbonyl-Diaminoalkanes (**6a–c**)

General Procedure to Synthesize *N*-*t*-Butoxy Carbonyl-Diaminoalkanes (**6a**,**b**)

In a 100 mL three-necked flask equipped with a magnetic stirrer, reflux condenser, drip funnel, and nitrogen valve, the proper diaminoalkane dissolved in dichloromethane (DCM) was introduced and, after cooling to −10 °C, a solution of di-*t*-butyl-di-carbonate (Boc_2_O) in DCM was dripped slowly (90 min). The reaction mixture was kept under stirring in a nitrogen atmosphere at room temperature until the reagents disappeared (TLC, eluent mixture: MeOH/ethyl acetate (EtOAc) 1/1, coloring reagent: 2% ninhydrin in EtOH). The solvent was removed under reduced pressure, and the obtained product was taken up with H_2_O (30 mL) and filtered, obtaining white solids which were identified as the side *bis*-protected products *N*, *N*’-(bis-*t*-butoxy carbonyl)-diaminoalkanes (**7a**,**b**). After saturation with NaCl, the filtrates were extracted with EtOAc (3 × 30 mL), and the organic extracts were dried over Na_2_SO_4_ overnight. By removal of the solvent at reduced pressure, the desired *N*-*t*-Butoxy carbonyl-diaminoalkanes (**6a**,**b**) were achieved in the form of light-yellow oils. The detailed experimental data have been included in Appendix A.

Procedure to Synthesize *N*-*t*-Butoxycarbonyl-1,6-diaminohexane (**6c**)

In a 100 mL three-necked flask equipped with a magnetic stirrer, reflux condenser, drip funnel, and nitrogen valve, 1,6-diamminohexane (8.45 g, 72.72 mmol) and dioxane (25 mL) were introduced. The solution was added dropwise with *S*-*t*-butoxycarbonyl-4,6-dimethyl-2-mercaptopyrimidine (**8**) (5.30 g, 22.03 mmol) dissolved in dioxane (21 mL), then left stirring at room temperature overnight. The thus obtained suspension was filtered to remove the solid 4,6-dimethyl-2-mercaptopyrimidine (**9**) (4.40 g, 31.38 mmol), concentrated to approximately 20 mL, and added with 40 mL of water. The new precipitate was filtered, washed with Et_2_O, and dried under reduced pressure. The obtained solid was identified as *N*, *N’*-(*bis*-*t*-butoxy carbonyl)-1,6-diaminohexane (**7c**) (0.5089 g, 1.61 mmol, yield 7.3%). The filtrate was concentrated under reduced pressure to eliminate dioxane, saturated with NaCl (14 g), and extracted with EtOAc (4 × 18 mL). Upon evaporation of the solvent, **6c** was achieved as oil, which was taken up with water (36 mL) and acidified with 12M HCl up to pH = 2–3. The solution was extracted with EtOAc, then saturated with NaCl for precipitating **6c** hydrochloride salt (**6c HCl**), which was filtered and brought to constant weight at reduced pressure (4.48 g, 17.72 mmol, yield 80.3%). Then, **6c HCl** was dissolved in water (20 mL), treated with a 10% NaOH solution (8 mL) until pH = 14, and extracted with CHCl_3_ (2 × 25 mL). The extracts were combined and dried on anhydrous MgSO_4_. After the elimination of the solvent, **6c** was provided as a colorless oil (3.07 g, 14.19 mmol, 80.1% yield).

Characterization of Compounds **6a–c** and **7a–c**

Compound **6a**. FTIR (film, cm^−1^): 3363 (NH), 2977, 2933, 2870 (CH_3_, CH_2_), 1695 (C=O). ^1^H NMR (CDCl_3_, 300MHz, δ ppm): 1.28 (s, 2H, NH_2_), 1.44 (s, 9H, CH_3_ Boc), 2.79 (t, 2H, CH_2_, *J* = 5.9 Hz), 3.16 (q, 2H, CH_2_, *J* = 5.8 Hz, *J* = 11.8 Hz), 5.49 (broad s, NH). ^13^C NMR (75.5 MHz, δ ppm): 28.43, 41.88, 43.41, 79.21, 156.27.

Compound **6b**. FTIR (film, cm^−1^): 3358 (NH), 2976, 2932, 2865 (CH_3_, CH_2_), 1705 (C=O). ^1^H NMR (CDCl_3_, 300 MHz, δ ppm): 1.17 (s, 2H, NH_2_), 1.44 (s, 9H, CH_3_ Boc), 1.46–1.54 (m, 4H, CH_2_CH_2_), 2.71 (t, 2H, CH_2_, *J* = 6.7 Hz), 3.11 (q, 2H, CH_2_, *J* = 6.1 Hz), 5.21 (broad s, NH).

Hydrochloride salt of compound **6c**. M. p. 162–164 °C (lit. [57] m. p.: 162.5–163 °C). FTIR (KBr, cm^−1^): 3378 (NH), 2983, 2936, 2866 (CH_3_, CH_2_), 1687 (C=O). ^1^H NMR (D_2_O, 300 MHz, δ ppm): 1.37 (m, 4H, 2 CH_2_), 1.43 (s, 9H, CH_3_ Boc), 1.58 (m, 4H, CH_2_CH_2_), 2.97–3.08 (m, 4H, 2 CH_2_). ^13^C NMR (75.5 MHz, δ ppm): 25.39, 25.54, 26.79, 27.97, 28.86, 39.66, 40.06, 80.89, 158.42.

Compound **6c**. FTIR (film, cm^−1^): 3344 (NH), 2976, 2930, 2857 (CH_3_, CH_2_), 1695 (C=O). ^1^H NMR (CDCl_3_, 300 MHz, δ ppm): 1.44 (s, 9H, CH_3_ Boc), 1.25–1.55 (m, 8H, 4 CH_2_ + s, 2H, NH_2_), 2.68 (t, 2H, CH_2_, *J* = 6.6 Hz), 3.10 (q, 2H, CH_2_, *J* = 6.2 Hz), 4.69 (broad s, NH). ^13^C NMR (75.5 MHz, δ ppm): 26.55, 26.64, 28.45, 30.09, 33.61, 40.53, 42.05, 78.97, 156.06.

Compound **7a**. M. p. 135–140 °C. FTIR (KBr, cm^−1^): 3382 (NH), 1684 (C=O). ^1^H NMR (CDCl_3_, 300 MHz, δ ppm): 1.44 (s, 18H, CH_3_ Boc), 3.22 (s, 4H, CH_2_NH), 5.19 (broad s, 2H, 2NH). ^13^C NMR (75.5 MHz, δ ppm): 28.42, 40.82, 79.33, 156.44.

Compound **7b** was not characterized.

Compound **7c**. FTIR (KBr, cm^−1^): 3442 (NH), 1687 (C=O). ^1^H NMR (CDCl_3_, 300 MHz, δ ppm): 1.32 (m, 4H, 2 CH_2_), 1.44 (s, 18H, CH_3_ Boc + m, 4H, 2 CH_2_), 3.10 (q, 4H, 2 CH_2_NH), 4.66 (broad s, 2H, 2NH). ^13^C NMR (75.5 MHz, δ ppm): 26.37, 28.45, 30.03, 40.43, 78.99, 156.07.

#### 3.3.2. Synthesis and Characterization of *N*-*t*-Butoxy Carbonyl-*N*’-Acryloyl-Diaminoalkanes (**10a–c**)

General Procedure to Synthesize *N-t*-Butoxy Carbonyl-*N*’-Acryloyl-Diaminoalkanes (**10a–c**)

In a 100 mL three-necked flask equipped with a magnetic stirrer, reflux condenser, dripping funnel, and nitrogen valve, acryloyl chloride dissolved in the proper solvent (CHCl_3_ for **6a** and **6b** and DCM for **6c**) was introduced. After cooling to −10 °C, triethylamine (TEA) and a solution of compounds **6a**, **6b,** or **6c** in the proper solvent, as reported above, were added slowly. The reaction mixture was kept under nitrogen and vigorous stirring at room temperature until the reagent disappeared (TLC, eluent mixture: MeOH/ethyl acetate 1/1, coloring reagent: 2% ninhydrin in EtOH). After the removal of the solvent under reduced pressure, the white solid (from **6a**), the oil (from **6b**), or the syrups (from **6c**) obtained were taken up with water (25, 30, or 25 mL), extracted with CHCl_3_ (3 × 30 mL) and the combined organic phases were dried over MgSO_4_ overnight. By removing the solvent at reduced pressure, **10a** and **10c** were provided in the form of white solids, while **10b** was first obtained as yellow oil, which was coagulated with hexane, achieving a yellow powdery solid after filtration. The experimental data of these preparations have been included in Appendix A.

Characterization of Compounds **10a–c**

Compound **10a**. M. p.: 95–96 °C (lit. [57] m. p.: 102.6–103 °C). FTIR (KBr, cm^−1^): 3304 (NH), 1696 (C=O), 1662 (C=O), 957, 991 (CH_2_=CH-). ^1^H NMR (CDCl_3_, 300 MHz, δ ppm): 1.43 (s, 9H, CH_3_ Boc), 3.31 (m, 2H, CH_2_), 3.44 (m, 2H, CH_2_), 5.19 (broad s, NH), 5.63 (dd, 1H di CH_2_=CH, *J gem* = 1.7 Hz, *J cis* = 10.1 Hz), 6.13 (dd, 1H, CH_2_=CH, *J cis* = 10.1 Hz, *J trans =* 17.0 Hz), 6.26 (dd, 1H di CH_2_=CH *J gem =* 1.5 Hz, *J trans =* 17.0 Hz), 6.74 (broad s, NH). ^13^C NMR (75.5 MHz, δ ppm): 28.38, 40.15, 40.91, 79.76, 126.23, 130.96, 157.13, 166.32.

Compound **10b**. M. p.: 88–90 °C. FTIR (KBr, cm^−1^): 3345 (NH), 3309 (NH), 1687 (C=O), 1661 (C=O). ^1^H NMR (CDCl_3_, 300 MHz, δ ppm): 1.44 (s, 9H, CH_3_ Boc), 1.51–1.60 (m, 4H, 2 CH_2_), 3.31 (m, 2H, CH_2_), 3.12 (m, 2H, CH_2_), 3.33 (q, 2H, CH2, *J* = 6.5 Hz), 4.81 (broad s, NH), 5.61 (dd, 1H di CH_2_=CH, *J gem* = 2.0 Hz, *J cis* = 9.8 Hz), 6.16 (dd, 1H, CH_2_=CH, *J cis* = 9.8 Hz, *J trans* = 17.0 Hz), 6.27 (dd, 1H di CH_2_=CH, *J gem* = 2.1 Hz, *J trans* = 17.0 Hz), 6.55 (broad s, NH). ^13^C NMR (75.5 MHz, δ ppm): 26.55, 27.70, 28,44, 39.23, 40.14, 79.24, 126.03, 131.09, 156.27, 165.84.

Compound **10c**. M. p.: 105–108 °C (lit. [57] m. p.: 108–108.5 °C). FTIR (KBr, cm^−1^): 3343 (NH), 1679 (C=O), 1652 (C=O). ^1^H NMR (CDCl_3_, 300 MHz, δ ppm): 1.34 (m, 4H, 2 CH_2_), 1.44 (s, 9H, CH_3_ Boc), 1.44–1.54 (m, 4H, 2 CH_2_), 3.09 (q, 2H, *J* = 6.3 Hz, CH_2_NH), 3.30 (q, 2H, *J* = 6.8 Hz, *J* = 6.2 Hz, CH_2_NH), 4.82 (broad s, NH), 5.60 (dd, 1H di CH_2_=CH, *J gem* = 2.5 Hz, *J cis* = 9.4 Hz), 6.19 (dd, 1H, CH_2_=CH, *J cis* = 9.4 Hz, *J trans* = 17.0 Hz), 6.27 (dd, 1H di CH_2_=CH, *J gem* = 2.5 Hz, *J trans* = 17.0 Hz), 6.67 (broad s, NH). ^13^C NMR (75.5 MHz, δ ppm): 25.89, 26.26, 28.44, 29.33, 29.98, 39.23, 40.20, 79.05, 125.84, 131.22, 156.25, 165.84.

#### 3.3.3. Synthesis and Characterization of Acryloyl Monomers (*N*-Acryloyl-1,6-diaminoalkane Hydrochlorides) (**11a–c**)

General Procedure to Synthesize *N*-Acryloyl-1,6-diaminoalkane hydrochlorides) (**11a–c**)

Into a 100 mL three-necked flask equipped with a magnetic stirrer and a nitrogen valve, compounds **10a**, **10b**, or **10c** were introduced together with a 3 N HCl solution in anhydrous EtOAc (7.5 mL). The reaction mixture (pH = 1) was kept under nitrogen and stirred at room temperature until the reagents disappeared (TLC, eluent mixture: MeOH/ethyl acetate 1/1, coloring reagent: ninhydrin 2% in EtOH). Then, the solvent was removed under reduced pressure to provide compounds **11a** directly as white solid, **11b** as yellow oil (1.41 g), which was crystallized with Et_2_O as dark solid, and **11c** as solid with was further purified by treatment with Et_2_O. The detailed experimental data of these preparations have been included in Appendix A.

Characterization of Compounds **11a–c**

Compound **11a** [57]. FTIR (KBr, cm^−1^): 3243 (NH), 3066 (NH_3_^+^), 1656 (C=O). ^1^H NMR (CD_3_OD, 300 MHz, δ ppm): 3.09 (t, 2H, CH_2_N, *J* = 6.0 Hz), 3.53 (t, 2H, CH_2_N, *J* = 6.0 Hz), 5.72 (dd, 1H, CH_2_=CH, *J* = 6.71 Hz), 6.25 (m, 2H, residual vinyl H). ^13^C NMR (75.5 MHz, δ ppm): 38.85, 41.48, 128.13, 132.16, 169.90.

Compound **11b**. M. p.: 145–147 °C. From CH_3_CN, m. p. 140–145 °C. FTIR (KBr, cm^−1^): 3262 (NH), 3073 (NH_3_^+^), 1651 (C=O). ^1^H NMR (CD_3_OD/D_2_O 2/1, 300 MHz, δ ppm): 1.58–1.78 (m, 4H, 2 CH_2_), 3.03 (t, 2H, CH_2_N, *J* = 7.1 Hz), 3.32 (t, 2H, CH_2_N, *J* = 6.6 Hz), 5.75 (dd, 1H, CH_2_=CH, *J gem* = 2.4 Hz, *J cis* = 9.3 Hz), 6.25 (m, 2H, residual vinyl H). ^13^C NMR (75.5 MHz, δ ppm): 23.80, 25.13, 37.92, 38.56, 126.21, 129.60, 167.45.

Compound **11c**. M. p.: 165–168 °C (lit. [57] m. p.: 164.5–165 °C). FTIR (KBr, cm^−1^): 3306 (NH), 3053 (NH_3_^+^), 1653 (C=O). ^1^H NMR (D_2_O, 300 MHz, δ ppm): 1.39 (m, 4H, 2 CH_2_), 1.63 (m, 4H, 2 CH_2_), 3.01 (t, 2H, CH_2_N, *J* = 7.5 Hz), 3.28 (t, 2H, CH_2_N, *J* = 6.8 Hz), 5.76 (dd, 1H, CH_2_=CH, *J gem* = 1.8 Hz, *J cis* = 9.8 Hz), 6.24 (m, 2H, residual vinyl H). ^13^C NMR (75.5 MHz, δ ppm): 25.19, 25.47, 26.61, 28.02, 39.22, 39.45, 126.99, 130.11, 168.49.

#### 3.3.4. Preparation of DMAA Copolymers of Compounds **11a–c** (CP11a–c/DMAA)

General Procedure to Polymerize **10a** and **10b** as well as **11c** with DMAA

In a tailed test tube equipped with a magnetic stirrer and a screw cap with a silicone septum, monomers **10a**, **10b**, or **11c**, anhydrous MeOH, DMAA, and AIBN (1% by weight compared to the reagents) were introduced under nitrogen. Upon complete solubilization, the solutions were transferred via steel needle into a single-neck polymerization flask equipped with a screw cap and silicone septum. The needle was washed with anhydrous MeOH, and the solution was degassed by bubbling nitrogen for 15 min. Then, it was kept under stirring at 60 °C. At the end of the polymerization, when it was started from **10a** and **10b**, the viscous solution was diluted with MeOH (1 mL), precipitated in Et_2_O (100 mL), and left under stirring overnight, achieving a cloudy solution. Upon decantation, oils were obtained, which were purified through three dissolution/precipitation cycles in MeOH/Et_2_O and one dissolution/precipitation cycle in DCM/Et_2_O. The copolymers were brought to constant weight at reduced pressure in the form of white solids. Otherwise, when the reaction was started from **11c**, the viscous solution was precipitated in Et_2_O, observing the formation of a compact agglomerate, which was left stirring overnight in 150 mL of fresh ether. After the removal of the solvent, the copolymer was purified through two MeOH (8 mL)/Et_2_O (150 mL) dissolution/precipitation cycles, obtaining a compact and sticky white mass, which was left stirring overnight again on Et_2_O for coagulation. The copolymer was filtered and brought to constant weight at reduced pressure. Experimental details concerning all the copolymerization reactions performed in this study, starting from **10a**, **10b**, and **11c,** have been included in Appendix A.

Characterization of Copolymers

CP10a/DMAA: FTIR (KBr, cm^−1^): 1631 (C=O). ^1^H NMR (CDCl_3_, 300 MHz, δ ppm): 1.43 (*t*-butyl Boc), 2.90 (CH_3_ DMAA).

CP10b/DMAA: FTIR (KBr, cm^−1^): 1633 (C=O). ^1^H NMR (CDCl_3_, 300 MHz, δ ppm): 1.43 (*t*-butyl Boc), 2.91 (CH_3_ DMAA).

CP11c/DMAA: FTIR (KBr, cm^−1^): 1629 (C=O). ^1^H NMR (CD_3_OD, 300 MHz, δ ppm): 2.95 (CH_3_ DMAA).

Fractioning of Copolymers CP10a, b/DMAA

The selected copolymer was solubilized in DCM (2% solution) and transferred by filtering through the paper into a 250 mL three-neck flask equipped with a mechanical stirrer and a dripping funnel and thermostated at 25 °C. The clear colorless solution was added dropwise with Et_2_O at 25 °C under slow stirring until a cloudy mixture was obtained. After decantation, the supernatant was evaporated at reduced pressure, while the oils obtained were dissolved in DCM and precipitated in Et_2_O. The copolymer was filtered and brought to constant weight using a mechanical pump.

General Procedure to Deprotect *N*-Boc protected Copolymers CP10a, b/DMAA

The *N*-Boc protected CP10a/DMAA and CP10b/DMAA copolymers were introduced into a flask equipped with a magnetic stirrer and treated with a 3 N HCl solution in EtOAc at room temperature. The biphasic mixtures were kept under stirring for 4 h. Then, solvents were eliminated in a vacuum, obtaining sticky solids, which were treated twice with absolute EtOH and then twice with toluene, evaporating the solvent each time. The obtained copolymers appeared as white glassy solids, which were brought to constant weight at reduced pressure. Appendix A shows the data of the deprotections carried out in this study.

Characterization of Copolymers

CP11a/DMAA: FTIR (KBr, cm^−1^): 1625 (C=O). ^1^H NMR (CD_3_OD, 300 MHz, δ ppm): 2.92 (CH_3_ DMAA).

CP11b/DMAA: FTIR (KBr, cm^−1^): 1628 (C=O). ^1^H NMR (CD_3_OD, 300 MHz, δ ppm): 2.92 (CH_3_ DMAA).

### 3.4. Synthesis and Characterization of Methacrolein (MA) Copolymer with DMAA (CPMA/DMAA)

Methacrolein (100 mg, 1.43 mmol, 118 μL), DMAA (567.0 mg, 5.72 mmol, 585 μL), AIBN (6.7 mg, 1.0%), and DMF (2 mL) were introduced into a 10 mL tailed test tube, equipped with a magnetic stirrer and a screw cap with a silicone septum. The obtained solution was then transferred into a 10 mL balloon with a neck equipped with a screw cap and a silicone septum by a needle. After washing the needle with DMF (1 mL), the transferred mixture was degassed by bubbling nitrogen for 15 min and kept under stirring at 60 °C for 24 h. The viscous solution was precipitated in Et_2_O (75 mL) to provide the copolymer as a white solid (197.6 mg, 29.6%).

FTIR (KBr, cm^−1^): 3454 (NH), 1720 (C=O), 1630 (C=O). ^1^H NMR (CDCl_3_, 300 MHz, δ ppm): 2.90 (CH_3_ DMAA), 9.00–9.60 (CHO MA).

### 3.5. Enzymatic Oxidation Assays of the Prepared Copolymers

The copolymer under study was introduced into a 10 mL test tube equipped with a magnetic stirrer and was dissolved in the appropriate buffer system at 38 °C. Then, a previously prepared solution of the enzyme in the same buffer was added to the copolymer solution, leaving the mixture under stirring overnight. After 24 h, the Schiff test was performed using the Shiff reagent (Merck, Darmstadt, Germany) to highlight the formation of the carbonyl function. The essay was considered positive when a fuchsia coloration occurred. The results have been reported in Appendix A.

### 3.6. Particle Size, Zeta Potential (ζ-p), and Polydispersity Index (PDI) of Selected Copolymers

The hydrodynamic size (diameter) (Z-AVE, nm) and polydispersity index (PDI) of copolymers selected on the basis of enzymatic oxidation results (Schiff test positive) were determined using Dynamic Light Scattering (DLS) analysis. Z-Ave and PDI measurements were performed in water mQ as a medium at a max concentration of copolymer of 3 mg/mL (pH = 7.4) in batch mode using a low-volume quartz cuvette (pathlength, 10 mm). The analysis was performed by a photon correlation spectroscopy (PCS) assembly equipped with a 50 mW He-Ne laser (532 nm) and thermo-regulated at the physiological temperature of 37 °C. The scattering angle was fixed at 90°. The results were the combination of three 10-min runs for a total accumulation correlation function (ACF) time of 30 min. The hydrodynamic particle size result was intensity-weighted and reported as the mean of three measurements ± SD. PDI value was reported as the mean of three measurements ± SD made by the instrument on the sample. The ζ-p was measured at 37 °C in mQ water as a medium, and an applied voltage of 100 V was used. The samples were loaded into pre-rinsed folded capillary cells, and twelve measurements were performed. The results have been reported in Appendix A.

### 3.7. Gelatine Crosslinking Tests

About 500 mg of Gel B (Merck, Darmstadt, Germany) and the appropriate buffer system (Appendix A) were introduced into a 50 mL container (Ø ext 35 mm, Ø int 28 mm, h = 10 cm) under heating at 40 °C until the gelatine was completely dissolved (about 30 min). Solutions in the same buffer system of the copolymer under study and of the selected amine oxidase (LO or PAO) were added to the Gel B solution. The mixture was left under heating at 40 °C for 24 h. After this time, the reaction mixture was subjected to dialysis. In a CelluSep T3 membrane tube having a molecular weight cut-off in the range of 12,000–14,000. Dialysis was performed to eliminate urea, other low molecular weight species, and any unreacted copolymer.

The membrane was immersed in mQ H_2_O and left to stir for 24 h. The dialyzed solution was transferred to a 500 mL beaker, and the membrane was washed with mQ H_2_O; then, all the dialyzed material and washings were moved into two Petri dishes (Ø = 10 cm) for evaporation of the solvent at 40 °C. The dried gelatine was detached from the capsules with the help of THF and extracted with water at 40 °C or DMSO at 70 °C in the case of the CPMA/DMAA samples for 24 h. The residual solvent was removed either using a desiccator on P_2_O_5_ or by evaporation on a Petri dish at 40 °C. The data and results concerning each experiment carried out specifically are available in Appendix A.

### 3.8. Titration of Gelatine and Crosslinked Gelatine

#### 3.8.1. Acid/Base Titrations of Gelatine

About 220–250 mg of Gel B or crosslinked Gel B and 15 mL of mQ H_2_O were introduced into a 50 mL beaker equipped with a magnetic stirrer and left stirring for 25 min, thus allowing the gelatine to swell and making the groups titratable better accessible. The titration was carried out with a standard solution of NaOH 0.1033 N, adding aliquots of 0.06 mL and measuring the pH values. The experiments were carried out in triplicate, and results were expressed as mean of three independent measures ± standard deviation. Titration data were reported in the graph (pH vs. NaOH (mL)), obtaining the titration curves, from which the volume of NaOH added at pH values equal to 8.50 and 11.50 were extrapolated. The quantity of protons released from the sample (N_H+_) expressed in mmol/g of gelatine was calculated using Equation (1).
N_H+_ = [(V _NaOH_ ∙ C _NaOH_)/g _gel B_]∙[1 − 10 ^(pH g − pH w)^](1)

In Equation (1), V _NaOH_ and C _NaOH_ are the volume and concentration of the titrant, g is the grams of gelatine sample that was titrated, while pH g and pH w are the pH values of the gelatine solution and of water after adding the volume V _NaOH_, respectively. The results of the titrations of Gel B are available in Section 2.5.1, Appendix A, while those of crosslinked Gel B are available in Appendix A.

Using data from these titrations, the percentage of crosslinking of the reticulated samples was calculated according to Equation (2).
CL% = {[(mmol NH_2_) **_Gel B_** − (mmol NH_2_) **_Gel B CL_**]/(mmol NH_2_) **_Gel B_**} × 100(2)

In Equation (2), CL means crosslinked. The results have been included in Appendix A.

#### 3.8.2. UV Titrations of Gelatine

UV titrations of gelatine were carried out following a procedure described in the literature [63]: A sample of pristine Gel B or crosslinked Gel B (approximately 11 mg), 1 mL of a 4% NaHCO_3_ solution, and 1 mL of a 0.5% TNBS (2,4,6-trinitrobenzenesulfonic acid) solution were introduced into a 50 mL test tube. The thus obtained orange suspension was heated in an oil bath at 40 °C for 4 h. Then, 3 mL of a 6 M HCl solution was added, observing effervescence and the appearance of a yellow coloration. The mixture was then heated to 116 °C for 1 h. The solution was transferred into a 100 mL separating funnel; the test tube was washed with 5 mL of mQ H_2_O and extracted with Et_2_O (3 × 4 mL). The aqueous phase was heated in a water bath for 15 min to eliminate traces of residual ether. 5 mL of aqueous phase was then taken with a single-notched pipette and diluted to 20 mL in a flask, using mQ H_2_O. A solution sample prepared under the same conditions but using the pristine Gel B (11.2 mg) was used as blank. UV analysis was performed at 346 nm. The experiments were carried out in triplicate, and results were expressed as mean of three independent measures ± standard deviation (SD).

The content of the NH_2_ groups was calculated using Equation (3) [63].
Mol NH_2_ = (2A × 0.020)/(1.46 × 10^4^ × g)(3)

In Equation (3), A was the measured absorbance, 1.46 was the ε of the chromophore at 346 nm, and g was the weight of Gel B used. The results of the titrations of Gel B are available in Section 2.5.6, Appendix A, while the results obtained for the crosslinked Gel B are available in Appendix A.

Using data from these titrations, the percentage of crosslinking of the reticulated samples was calculated according to Equation (3).

### 3.9. Cytotoxicity Experiments

The human neuroblastoma cell lines IMR-32 and SH SY5Y were maintained in a complete medium (Dulbecco’s modified Eagle medium; Sigma, Milan, Italy) containing 10% *v*/*v* heat-inactivated fetal bovine serum (Gibco-Invitrogen S.r.l., Carlsbad, CA, USA) and 50 IU/mL penicillin G; 50 µg/mL streptomycin sulfate; and 2 mM L-glutamine (all reagents from Euroclone S.p.A., Milan, Italy). Cells were periodically tested for mycoplasma contamination (Mycoplasma Reagent Set, Aurogene s.p.a, Pavia, Italy). To assay cell proliferation, cells were seeded in triplicate in a 96 w plate at 3000 to 10,000 cells per well in 200 µL of complete medium. After 24 h, the medium was changed, and the cells were exposed for 24, 48, or 72 h to P5 at 0.1, 0.5, 1, 2, 5, 7.5, 10, and 15 µM. The effect on cell growth was evaluated by a fluorescence-based proliferation and cytotoxicity assay (CyQUANT^®^ Direct Cell Proliferation Assay, Thermo Fisher Scientific, Life Technologies, Monza Brianza, Italy) according to the manufacturer’s instructions. Briefly, at the selected times, an equal volume of detection reagent was added to the cells in culture and incubated for 60 min at 37 °C. The fluorescence of the samples was measured using the monochromator-based M200 plate reader (Tecan, Männedorf, Switzerland) set at 480/535 nm.

### 3.10. Statistical Analysis

All the experiments were performed at least three times. Each set of experimental conditions for the biological assays was tested in 96-well plates and carried out in triplicate. Differential findings among the experimental groups were determined by two-way ANOVA (analysis of variance) with Bonferroni post-tests using GraphPad Prism 8 (GraphPad Software v8.0, San Diego, CA, USA). Asterisks indicate the following *p*-value ranges: * = *p* < 0.05, ** = *p* < 0.01, *** = *p* < 0.001.

## 4. Conclusions

The main scope of the present study was the preparation of new materials with the potential to be employed in TE, which is a critical step when a TE study is undertaken. Starting from reported studies demonstrating that effective scaffolds for cell proliferation experiments could be obtained by the reinforcement of gelatine via crosslinking reactions, a versatile new protocol to crosslink gelatine B was developed here. It was based on the enzymatic oxidation of amine-containing copolymers, followed by crosslinking via imine bond formation and/or aldolic reactions, between gelatine and oxidized copolymers, as it occurs in nature for collagen.

Three types of monomers containing primary amine groups and their copolymers with DMAA were synthesized and characterized, which were demonstrated to be good to excellent substrates for LO and were used in successful crosslinking experiences.

By synthesizing a copolymer already containing aldehyde residues (MA/DMAA) and assumed as an LO-independent crosslinking agent, it demonstrated the importance of LO in the designed gelatine crosslinking processes, also in the presence of already oxidized groups.

Based on its percentage of crosslinking, the most suitable sample for further physicochemical and biological tests was obtained using the amine copolymer CP5/DMMA-43.1 as a crosslinking agent (M21). Sample M21 could be considered as an excellent material to develop a new scaffold for TE since, in cytotoxicity experiments carried out with the structural analogous of the synthetic copolymer used to prepare it, no significant reduction in cell viability but cell proliferation was observed.

## Data Availability

Data supporting reported results are included in this manuscript and in the related Appendix A.

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
