# Peer review of "Synthesis and Characterization of Amine and Aldehyde-Containing Copolymers for Enzymatic Crosslinking of Gelatine"

_ijms, 2024, doi:10.3390/ijms25052897_

Round 1

Reviewer 1 Report

Comments and Suggestions for Authors

I had the opportunity to review the manuscript entitled “Synthesis and Characterization of Amine and Aldehyde-Containing Copolymers for Enzymatic Crosslinking of Gelatine” submitted to Int. J. Mol. Sci. The authors present the preparation of an amino butyl styrene monomer (5) and its copolymerization with dimethylacrylamide and acrylic acid. Then, they present synthesis of three acryloyl amidoamine monomers and their copolymers with dimethylacrylamide. Finally, they prepared a methacrolein copolymer already possessing aldehyde groups. From oxidation tests they identified the best substrates (lysyl oxidase (LO)) were employed with methacrolein-based copolymers in crosslinking tests with gelatine. Finally, the most promising materials were chosen for further investigations into structure and mechanical properties, for carrying out biodegradability and cytotoxicity assays, as well as experiments of cell adhesion and proliferation.

 I think that the manuscript is way too long, and maybe a bit tiring for the reader. There are too many Schemes and Figures and some of them present similar things (e.g. Figure 2b / Scheme 4). The authors should try to rearrange the content to be less lengthy and stick more to the characterization discussion.

As far as the comment of the authors that the most promising materials (CP5/DMMA3.1 and the aldehyde type CPMA/DMAA-20.0) were chosen for further investigations into structure and mechanical properties, for carrying out biodegradability and cytotoxicity assays, as well as experiments of cell adhesion and proliferation: have the authors performed any experiments concerning biodegradation and cytotoxicity capability?

Other than that, I have to say that the authors have chosen representative bibliography and generally, the English language is fine.

Author Response

I had the opportunity to review the manuscript entitled “Synthesis and Characterization of Amine and Aldehyde-Containing Copolymers for Enzymatic Crosslinking of Gelatine” submitted to Int. J. Mol. Sci. The authors present the preparation of an amino butyl styrene monomer (5) and its copolymerization with dimethylacrylamide and acrylic acid. Then, they present synthesis of three acryloyl amidoamine monomers and their copolymers with dimethylacrylamide. Finally, they prepared a methacrolein copolymer already possessing aldehyde groups. From oxidation tests they identified the best substrates (lysyl oxidase (LO)) were employed with methacrolein-based copolymers in crosslinking tests with gelatine. Finally, the most promising materials were chosen for further investigations into structure and mechanical properties, for carrying out biodegradability and cytotoxicity assays, as well as experiments of cell adhesion and proliferation.

 I think that the manuscript is way too long, and maybe a bit tiring for the reader. There are too many Schemes and Figures and some of them present similar things (e.g. Figure 2b / Scheme 4). The authors should try to rearrange the content to be less lengthy and stick more to the characterization discussion.

We thank the Reviewer for his/her precious suggestions which enabled us to improve our work. The work has been completely reworked. Figures have been dramatically reduced from 23 to 8, either removing those useless or moving them to Supplementary Materials. The same has been made for Tables. Discussion has been shortened and repetitions have been removed.  

As far as the comment of the authors that the most promising materials (CP5/DMMA3.1 and the aldehyde type CPMA/DMAA-20.0) were chosen for further investigations into structure and mechanical properties, for carrying out biodegradability and cytotoxicity assays, as well as experiments of cell adhesion and proliferation: have the authors performed any experiments concerning biodegradation and cytotoxicity capability?

We thank the Reviewer for his answer. In fact, although something inherent this point was present already in the original manuscript (lines 1201-1203, revised manuscript), more clarifications were needed. In the revised version of the paper, both in the abstract (lines 27-30) and in the conclusions (lines 1229-1231), it has been explained that the mentioned experiments, are currently underway, and will be reported in our next work. The scope of the present paper was in fact to synthetize the polymeric crosslinking agents, detect among the amine ones the best substrates for LO, crosslinking gelatine B in different conditions and determine the degree of crosslinking of the obtained materials.

Other than that, I have to say that the authors have chosen representative bibliography and generally, the English language is fine.

We thank a lot the Reviewer for his/her positive comment.

Reviewer 2 Report

Comments and Suggestions for Authors

The article proposes a new protocol for crosslinking gelatine hydrogels to improve their mechanical properties. It is mainly based on the formation of imine bonds and/or the aldolic condensation reactions occurring between gelatine and certain synthesized copolymers. The main application of this proposal would be in the tissue engineering (TE) field.

I consider the topic interesting for the journal's potential readers, but after reading the article two times, it is still hard to link several parts. First, the experimental works described in the long section of Results and Discussion are not properly summarised. I do not see a clear structure in the results. The authors should rethink the article, select the data that is important to report, and, of course, dramatically reduce the length of the paper. It is mandatory to include ONLY the relevant data. Including more data is not a sympton of better results or better paper. In the reviewer´s opinion, the current format is more like a laboratory notebook than a research article. 

There are some parts that were also critical to deciding to reject the paper and not consider it for publication:

- 13C NMR anmd 1H NMR spectra figures are not acceptable. The current format is a "raw image" makes impossible to evaluate the results. Authors should edit properly and include figures "readable" or  comparable for any potential reader of the paper. 

- The same request is done for the FTIR spectra. It is mandatory to include refined figure, not the raw figures with numbers that are impossible to read. 

- Many parts of the results Section are repetitions of the "Materials and Methods". For example:

Line 569: "Gelatine used in this study was type B gelatine (Sigma-Aldrich, Darmstadt, Germany) hereinafter referred to as Gel B."

Line 594: "The content of the NH2 groups were calculated using Equation (2) [61]."

Line 603: "For the enzyme interaction and crosslinking assays, we used LO isolated by us from 603 bovine aorta [62] by applying only the first part of the protocol found in literature [63]. 604 The main steps of extraction process have been reported in Chart S1 in Supplementary 605 Materials. Concerning the preparate isolated by us, the SDS-PAGE electrophoretic analy- 606 sis evidenced the presence of LO in correspondence with the 32 KDa band. "

Line 702: Then the percentage of crosslinking of the samples M21, M31, M23, M26, M32 and M34 were calculated 703 according to Equation (3).

and many more... 

Comments on the Quality of English Language

Extensive editing still required, specially in the structure of the paper...

Author Response

The article proposes a new protocol for crosslinking gelatine hydrogels to improve their mechanical properties. It is mainly based on the formation of imine bonds and/or the aldolic condensation reactions occurring between gelatine and certain synthesized copolymers. The main application of this proposal would be in the tissue engineering (TE) field.

I consider the topic interesting for the journal's potential readers, but after reading the article two times, it is still hard to link several parts. First, the experimental works described in the long section of Results and Discussion are not properly summarised. I do not see a clear structure in the results. The authors should rethink the article, select the data that is important to report, and, of course, dramatically reduce the length of the paper. It is mandatory to include ONLY the relevant data. Including more data is not a sympton of better results or better paper. In the reviewer´s opinion, the current format is more like a laboratory notebook than a research article. 

We thank the Reviewer for his/her precious suggestions which enabled us to improve our work. The work has been completely reworked. Figures have been reduced from 23 to 8, either removing those useless or moving them to Supplementary Materials. The same has been made for Tables. Discussion has been shortened and repetitions have been removed, as well as non-relevant data.  

There are some parts that were also critical to deciding to reject the paper and not consider it for publication:

- 13C NMR anmd 1H NMR spectra figures are not acceptable. The current format is a "raw image" makes impossible to evaluate the results. Authors should edit properly and include figures "readable" or comparable for any potential reader of the paper. 

All NMR spectra images have been improved and now are clearly readable.

- The same request is done for the FTIR spectra. It is mandatory to include refined figure, not the raw figures with numbers that are impossible to read. 

All FTIR spectra images have been improved and now all numbers are clearly readable.

- Many parts of the results Section are repetitions of the "Materials and Methods". For example:

Line 569: "Gelatine used in this study was type B gelatine (Sigma-Aldrich, Darmstadt, Germany) hereinafter referred to as Gel B."

We apologize in advance to the Reviewer, but we have not found the signaled sentence in Materials and Methods section. Anyway, to satisfy the Reviewer the sentence has been removed.

Line 594: "The content of the NH2 groups were calculated using Equation (2) [61]."

We apologize in advance to the Reviewer, but we have not found the signaled sentence in Materials and Methods section. Anyway, to satisfy the Reviewer the sentence has been removed and equations (1) and (2) have been moved in Materials and Methods section.

Line 603: "For the enzyme interaction and crosslinking assays, we used LO isolated by us from 603 bovine aorta [62] by applying only the first part of the protocol found in literature [63]. 604 The main steps of extraction process have been reported in Chart S1 in Supplementary 605 Materials. Concerning the preparate isolated by us, the SDS-PAGE electrophoretic analy- 606 sis evidenced the presence of LO in correspondence with the 32 KDa band. "

We apologize in advance to the Reviewer, but we have not found the signaled paragraph in Materials and Methods section. The paragraph has been shortened but maintained.

Line 702: Then the percentage of crosslinking of the samples M21, M31, M23, M26, M32 and M34 were calculated 703 according to Equation (3).

and many more... 

We apologize in advance to the Reviewer, but we have not found the signaled sentence in Materials and Methods section. Anyway, to satisfy the Reviewer it has been removed, and equation (3) has been moved in Materials and Methods section.

Comments on the Quality of English Language

Extensive editing still required, specially in the structure of the paper...

The manuscript has been revised by our colleague Professor Deirdre Kantz, English mother tongue and teaching English at the University of Genoa and Pavia.

Round 2

Reviewer 2 Report

Comments and Suggestions for Authors

I finished the review of the paper entitled “Synthesis and Characterization of Amine and Aldehyde-Containing Copolymers for Enzymatic Crosslinking of Gelatine", and I have to say that the authors did a great job reducing the number of pages to enhance the interpretation of the results. Now that I can easily assess the proposed paper, I consider that par of the experimental work that the authors claimed they were undergoing is mandatory for publication (lines 26-27): "Experiments to investigate their structure, mechanical properties, biodegradability and cytotoxicity, as well as tests of cell adhesion and proliferation, are currently underway." At least, cytotoxicity should be discarded from the material proposed... I cannot see the relevance of the work without collecting this data. That is the main reason why I will ask for a major review.

Another important issue is the Introduction. I think that the paper still needs some changes before it is accepted for publication. I would like to propose the following ones in the Introduction Section:

- Figure 1: "Cells isolation and multiplication" change to Cells isolation and expansion"

- Reference [11] from Atala´s lab is not a recent work in TE, please review the recent literature in TE to show the "current picture".

- The text mentioned several "biodegradable polymers" as "more attractive". However, this is not correct if we just read the current literature. I would suggest focusing the introduction  to resorbable polymers. See current literature in:

(1) https://doi.org/10.1016/j.mtchem.2023.101818

- The authors defended the use of Gelatin but lines 97 - 99: "Unfortunately, collagen and gelatine possess modest mechanical properties, which limit their use in the preparation of efficient hydrogel scaffolds for TE, if not properly modified." have no sense to me. Both can create hydrogels, however, it is the stiffness that is not comparable with the ECM. Please review this issue.

- Lines 75 - 79: "Non-biodegradable polymers such as polymethyl methacrylate (PMMA) and poly-hydroxyethyl methacrylate (PHEMA) have contraindications for effects, including inflammation, mainly in long-term implants addressed through chemical modifications by introducing dextran segments [18,19]." Both references do not apply to long-term implants... Also, the publication dates are 2003 and 2000. The authors should review this and, in general, the Introduction to demonstrate that they understand the current state of the art of polymers in the field of TE.

Lastly, the authors should clarify the main findings or ideas included in the Conclusions Section. As far as I can read, the first paragraph is a "future works"..., and then the authors repeat the abstract in terms of what they did during the experimental work. But, what about conclusions? In addition, the sentence "Experiments to investigate their structure, mechanical properties, biodegradability, and cytotoxicity, as well as studies of cell adhesion and proliferation are currently underway." is repeated.

Comments on the Quality of English Language

Some paragraphs are still not clear after a second reading...

Author Response

I finished the review of the paper entitled “Synthesis and Characterization of Amine and Aldehyde-Containing Copolymers for Enzymatic Crosslinking of Gelatine", and I have to say that the authors did a great job reducing the number of pages to enhance the interpretation of the results. Now that I can easily assess the proposed paper, I consider that par of the experimental work that the authors claimed they were undergoing is mandatory for publication (lines 26-27): "Experiments to investigate their structure, mechanical properties, biodegradability and cytotoxicity, as well as tests of cell adhesion and proliferation, are currently underway." At least, cytotoxicity should be discarded from the material proposed... I cannot see the relevance of the work without collecting this data. That is the main reason why I will ask for a major review.

We thank the Reviewer for having appreciated our revision work and for his/her comments and suggestions. Anyway, we do not fully agree with him/her (and for this we apologize to him/her) concerning the need to include cytotoxicity experiments to make the present work relevant. The scope of the present work was developing a new and versatile protocol to crosslink gelatin and obtain materials promising to be used as scaffolds for tissue engineering (TE). To this end, detecting in the natural reinforcement of collagen by its LO-mediated crosslinking, a good strategy for our objective, we have performed a laborious work of synthesis and characterization of some monomers and their copolymers, as possible substrate of LO, due to their amine groups content, mimicking the lysine residues of collagen. The most part of prepared copolymers demonstrated to be from good to excellent substrates for LO and were successful in experiment of gelatine crosslinking, thus providing materials with good percentage of reticulation, and therefore with potentialities for being further developed as scaffolds for TE. Biological experiments were not in the aims of this work, which, as the same Reviewer has evidenced, is already sufficiently long and complex. Additionally, examples exist in literature of works finalized to prepare materials promising for regenerative medicine, which lack biological evaluations (https://doi.org/10.1109/EMBC48229.2022.9871508, https://doi.org/10.1007/s10853-016-9747-4). Anyway, since among the prepared crosslinked materials, the most promising one (M21, 70.5% crosslinking) was that obtained using copolymer CP5/DMMA, structural analogous of a previously studied copolymer P5, to demonstrate the biocompatibility of M21, we have provided the cytotoxicity results previously obtained with P5 on two cell lines. To this end, such results have been discussed in Section 2 (lines 629-655) and the related experimental procedure has been reported in Section 3 (lines 1003-1025).

Another important issue is the Introduction. I think that the paper still needs some changes before it is accepted for publication. I would like to propose the following ones in the Introduction Section:

- Figure 1: "Cells isolation and multiplication" change to Cells isolation and expansion"

We thank the Reviewer for his/her comment which has been addressed. Please, see the revised Figure 1.

- Reference [11] from Atala´s lab is not a recent work in TE, please review the recent literature in TE to show the "current picture".

The reviewer is right. The recent literature on TE has been reviewed and more recent achievements have been included in the revised manuscript with additional references (lines 59-64).

- The text mentioned several "biodegradable polymers" as "more attractive". However, this is not correct if we just read the current literature. I would suggest focusing the introduction  to resorbable polymers. See current literature in:

  • https://doi.org/10.1016/j.mtchem.2023.101818

On suggestion of the Reviewer, we have considered Ref. https://doi.org/10.1016/j.mtchem.2023.101818, and have cited it in our revised manuscript (Ref. 14, revised). Additionally, we have focused our attention on bio-resorbable, bio-erodible and bio-adsorbable polymers, describing their characteristics according to a recent paper (Ref. 25, revised manuscript). Please, see lines 102-111.

- The authors defended the use of Gelatin but lines 97 - 99: "Unfortunately, collagen and gelatine possess modest mechanical properties, which limit their use in the preparation of efficient hydrogel scaffolds for TE, if not properly modified." have no sense to me. Both can create hydrogels, however, it is the stiffness that is not comparable with the ECM. Please review this issue.

On the precious suggestion of the Reviewer, the issue has been reviewed. Please, see lines 127-130.

- Lines 75 - 79: "Non-biodegradable polymers such as polymethyl methacrylate (PMMA) and poly-hydroxyethyl methacrylate (PHEMA) have contraindications for effects, including inflammation, mainly in long-term implants addressed through chemical modifications by introducing dextran segments [18,19]." Both references do not apply to long-term implants... Also, the publication dates are 2003 and 2000. The authors should review this and, in general, the Introduction to demonstrate that they understand the current state of the art of polymers in the field of TE.

On the precious suggestion of the Reviewer, the sentence has been reformulated. Obsolete references have been replaced with recent ones on long-term side effects of PMMA. Please, see lines 85-94.

Lastly, the authors should clarify the main findings or ideas included in the Conclusions Section. As far as I can read, the first paragraph is a "future works"..., and then the authors repeat the abstract in terms of what they did during the experimental work. But, what about conclusions? In addition, the sentence "Experiments to investigate their structure, mechanical properties, biodegradability, and cytotoxicity, as well as studies of cell adhesion and proliferation are currently underway." is repeated.

On the suggestion of the Reviewer, repetitions in conclusions were removed, and conclusions were fully reformulated. Please see lines 1027-1069.

Comments on the Quality of English Language

Some paragraphs are still not clear after a second reading...

The manuscript has been newly double-checked to improve the quality of English language and to make it more easily readable and flowable.

Round 3

Reviewer 2 Report

Comments and Suggestions for Authors

The authors did a great job addressing most of reviewer´s comments, and the paper has significantly been enhanced. 

Comments on the Quality of English Language

No comments